# Image Manipulation via Neuro-Symbolic Networks

**Harman Singh**[1*]  **Poorva Garg**[1*†]  **Mohit Gupta**[1*†]  **Kevin Shah**[1]
**Arnab Kumar Mondal**[1]  **Dinesh Khandelwal**[2]
**Parag Singla**[1]  **Dinesh Garg**[2]
[1]Indian Institute of Technology, Delhi    [2]IBM Research AI
{harmansingh.iitd, poorva98, contact.mg24}@gmail.com
{kevin.shah.cs119, anz188380, parags}@iitd.ac.in
{dikhand1, garg.dinesh}@in.ibm.com

## Abstract

*Image manipulation via natural language text* – an extremely useful task for multiple AI applications but requires complex reasoning over multi-modal spaces. *Neuro-symbolic approaches* has been quite effective in solving such tasks as they offer better *modularity*, *interpretability*, and *generalizability*. A noteworthy such approach is NSCL [10] developed for the task of Visual Question Answering (VQA). We extend NSCL for the image manipulation task and propose a solution referred to as NEUROSIM. Unlike previous works, which either require *supervised data training* or can only deal with simple reasoning instructions over single object scenes; NEUROSIM can perform *complex multi-hop reasoning* over *multi-object scenes* and requires only *weak supervision* in the form of annotated data for the VQA task. On the language side, NEUROSIM contains neural modules that parse an instruction into a symbolic program over a Domain Specific Language (DSL) comprising manipulation operations that guide the manipulation. On the perceptual side, NEUROSIM contains neural modules which first generate a *scene graph* of the input image and then change the scene graph representation following the parsed instruction. To train these modules, we design *novel loss functions* capable of testing the correctness of manipulated object and scene graph representations via query networks. An image decoder is trained to render the final image from the manipulated scene graph representation. Extensive experiments demonstrate that NEUROSIM is highly competitive with state-of-the-art supervised baselines.

## 1 Introduction

In this paper, our aim is to build neuro-symbolic models for the task of *weakly supervised manipulation of images comprising multiple objects, via complex multi-hop natural language instructions.* We are interested in a weakly supervised solution that does not require explicit annotation in the form of manipulated images. We rely on the key intuition that this task can be achieved simply by querying the manipulated representation without ever explicitly looking at the target image.

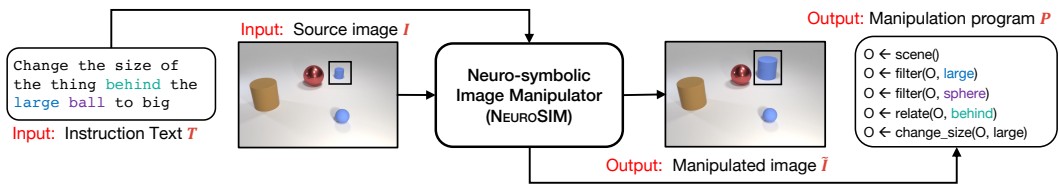

Figure 1: The problem setup.

---

*Work done while at IIT Delhi

†Equal Contribution

Neuro Causal and Symbolic AI Workshop at the 36th Conference on Neural Information Processing Systems (NeurIPS 2022).

Our solution builds on the work by Mao et al.[10] on Neuro-Symbolic Concept Learner (NSCL). We extend this work to incorporate the notion of manipulation of a given image. At high level, our solution works as follows – i) We first introduce a manipulation operation as part of the DSL and then learn to parse the natural language manipulation instruction comprising these concepts via a hierarchical parser [2]. The parsed instruction is represented in the form of a symbolic program dictating the manipulation to be performed on the input image. ii) We make use of a pre-trained network to extract latent object representations. iii) We make use of concept quantization networks to ground the linguistic concepts into visual objects and their attributes. iv) As one of our main contributions, we design neural modules that given a manipulation operator, transform the object (scene) so as to have the characteristics (attributes) as dictated by the parsed manipulation instruction. These neural modules are trained using *novel loss functions* that measure the faithfulness of the manipulated scene and object representations by accessing a separate set of *query networks* that are trained using VQA annotations. v) Separately, a network is trained to render the image from a scene graph representation using a combination of $L_1$ and adversarial losses in the style of [9]. We refer to our system as Neuro-Symbolic Image Manipulator (NEUROSIM). Figure 1 shows an example of (input,output) pair for NEUROSIM. For our experiment purposes, we extend CLEVR, a benchmark dataset for VQA, to incorporate manipulation instructions and create a dataset referred to as *Complex Image Manipulation via Natural Language Instructions (*CIM-NLI*)*. Our evaluation on CIM-NLI dataset shows that we are highly competitive with state-of-the-art supervised approaches [13, 3] for this task, and specifically performing well on instructions which involve multi-hop reasoning.

## 2   Related Work

In terms of task complexity, the closest to us are approaches such as TIM-GAN [13], GeNeVA [3], which build an encoder decoder architecture and work with a latent representation of the image as well as the manipulation instruction. They require explicit annotations in terms of manipulated images during training. We argue that this can require a significant more annotation effort, compared to our weak supervision setting, where we only need visual question answer annotations. Unlike us, these approaches work with purely neural models, and as shown in our experiments, their performance is heavily dependent on the amount of data available for training.

In terms of technique, the closest to our work are neuro-symbolic approaches for Visual Question Answering (VQA) such as  NSVQA [12], NSCL [10], Neural Module Networks [1] and its extensions [6, 7]. While these works' modeling approach is similar to ours and consists of constructing latent programs, the desired tasks are different. They solve the task of VQA, whereas our goal is to solve the task of automated image manipulation.

## 3   NEUROSIM: Neuro-Symbolic Image Manipulator

Figure 2 captures a high level architecture of the proposed NEUROSIM pipeline. NEUROSIM allows editing images containing multiple objects, via complex natural language instructions. Similar to Mao et al. [10], NEUROSIM assumes the availability of a *domain-specific language* (DSL) for parsing the instruction text $T$ into an executable program $P$. It reasons over the image for locating where the manipulation needs to take place followed by carrying out the manipulation operation. The first three modules, namely *i) visual representation network, ii) semantic parser,* and *iii) concept quantization network* are borrowed from the NSCL pipeline [10] and suitably customized/trained as required for our purpose. In what follows, we describe design as well as training mechanism of NEUROSIM.

### 3.1   Modules Inherited from NSCL

**1) Visual Representation Network:** Given input image $I$, this network converts it into a scene graph $G_I = (N, E)$. The nodes $N$ of this scene graph are object embeddings and the edges $E$ are embeddings capturing relationship between pair of objects (nodes). Node embeddings are obtained by passing the bounding box of each object (along with the full image) through a ResNet-34 [4]. Edge embeddings are obtained by concatenating the corresponding object embeddings.
**2) Semantic Parsing Module:** The input to this module is a manipulation instruction text $T$ (or a question when training on VQA task) in natural language. Output is a *symbolic program $P$* generated by parsing the input text. We extend the DSL used by [10] for incorporating manipulation operators.

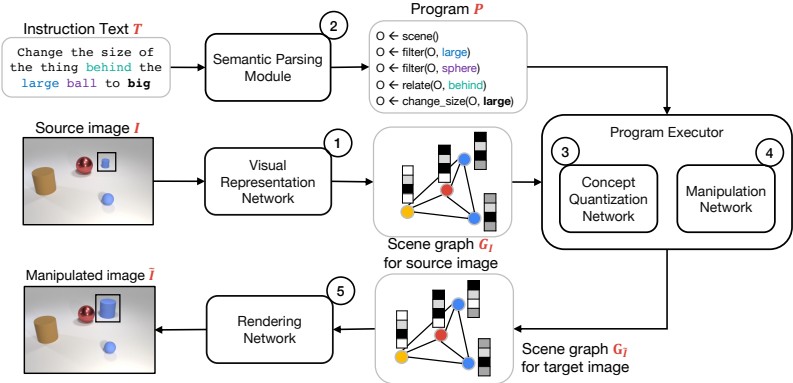

Figure 2: High level architecture of NEUROSIM.

**3) Concept Quantization Network:** Following [10], any object in an image is defined by the set of *visual attributes* $(A)$, and set of symbolic values $(S_a)$ for each attribute $a \in A$. For example, attributes can be *shape, size,* etc. Different symbolic values allowed for an attribute are also known as *concepts*. For example, $S_{\text{color}} = \{\text{red, blue, green}, \ldots\}$. Each visual attribute $a \in A$ is implemented via a separate neural network $f_a(\cdot)$ which takes the object embedding as input and outputs the attribute value for the object in a *continuous (not symbolic) space*. For example, let $f_{\text{color}} : \mathbb{R}^{d_{\text{obj}}} \to \mathbb{R}^{d_{\text{attr}}}$ represent a neural network for the *color* attribute and consider $o \in \mathbb{R}^{d_{\text{obj}}}$ as the object embedding. Then, $v_{\text{color}} = f_{\text{color}}(o) \in \mathbb{R}^{d_{\text{attr}}}$ is the embedding for the object $o$ pertaining to the color attribute. Each symbolic concept $s \in S_a$ for a particular attribute $a$ (for example, different kinds of colors) is also assigned a respective embedding in the same continuous space $\mathbb{R}^{d_{\text{attr}}}$. Such an embedding is denoted by $c_s$. These concept embeddings are initialized at random, and later on fine tuned during training. An attribute embedding ($v_{\text{color}}$ in the example above) can then be compared with the embeddings of all the concepts (for example, $c_{\text{red}}, c_{\text{blue}}$, etc.) using cosine similarity, for the purpose of various tasks such as filtering objects based on say, a specific color.

**Training:** As a first step of training NEUROSIM, we train all the above three modules via a three step curriculum learning process as outlined in [10]. During this training, semantic parser is trained jointly with the concept quantization networks for generating programs corresponding to the question texts coming from the VQA dataset. The corresponding output programs are composed of primitive operations coming from VQA DSL used in [10] (e.g. *filter*, *count*, etc.) and does not include constructs related to the manipulation. We assume that by now, first three modules have gotten trained for high accuracy on the VQA task. We refer the reader to [10] for more details.

### 3.2 Novel Modules and Training Procedure for NEUROSIM

NEUROSIM training starts with three sub-modules trained on the VQA task as described in Section 3.1. Next, we extend the original DSL to include an additional functional sub-module within semantic parsing module, *change*. Refer to appendix section A for details on the DSL. We now reset the semantic parsing module and train it again from scratch for generating programs corresponding to *image manipulation instruction text* $T$. Such a program is subsequently used by the downstream pipeline to reason over the scene graph $G_I$ and manipulate the image. In this step, the semantic parser is trained using an off-policy program search based REINFORCE [11] algorithm. Unlike the training of semantic parser for the VQA task, in this step, we *do not* have any final *answer like* reward supervision for training. Hence, we resort to a weaker form of supervision. In particular, consider an input instruction text $T$ and set of all possible manipulation program templates $\mathbb{P}_t$ from which one can create any actual program $P$ that is executable over the scene graph of the input image. For a program $P \in \mathbb{P}_t$, our reward is positive if this program $P$ selects any object (or part of the scene graph) to be sent to the change manipulation network. Once semantic parser is retrained, we clamp the first three modules and continue using them for the purpose of parsing instructions and converting images into their scene graph representations. Scene graphs are manipulated using our novel module called *change manipulation network* which is describe next.

**4) Change Manipulation Network:** For each visual attribute $a \in A$ (e.g. *shape, size,* ...), we have a separate *change neural network* that takes the pair of *(object embedding, embedding of the*

*changed concept)* as input and outputs the embedding of the *changed* object. For example, let $g_{\text{color}} : \mathbb{R}^{d_{\text{obj}}+d_{\text{attr}}} \rightarrow \mathbb{R}^{d_{\text{obj}}}$ represent the neural network that changes the *color* of an object. Consider $o \in \mathbb{R}^{d_{\text{obj}}}$ as the object embedding and $c_{\text{red}} \in \mathbb{R}^{d_{\text{attr}}}$ as the concept embedding for the *red* color, then $\widetilde{o} = g_{\text{color}}(o; c_{\text{red}}) \in \mathbb{R}^{d_{\text{obj}}}$ represents the changed object embedding, whose color would be *red*. After applying the change neural network, we obtain the changed representation of the object $\widetilde{o} = g_a(o; c_{s_a^*})$, where $s_a^*$ is the desired changed value for the attribute $a$. This network is trained using following losses.

$$\ell_a = -\sum\nolimits_{\forall s \in S_a} \mathbb{I}_{s=s_a^*} \ \log\left[p(h_a\left(\widetilde{o}\right) = s)\right] \tag{1}$$

$$\ell_{\overline{a}} = -\sum\nolimits_{\forall a' \in A, a' \neq a} \sum\nolimits_{\forall s \in S_{a'}} p(h_{a'}(o) = s) \log\left[p(h_{a'}(\widetilde{o}) = s)\right] \tag{2}$$

where, $h_a(x)$ gives the concept value of the attribute $a$ (in symbolic form $s \in S_a$) for the object $x$. The quantity $p\left(h_a(x) = s\right)$ denotes the probability that the concept value of the attribute $a$ for the object $x$ is equal to $s$ and is given as follows $p\left(h_a(x) = s\right) = \exp^{dist(f_a(x), c_s)} / \sum_{\widetilde{s} \in S_a} \exp^{dist(f_a(x), c_{\widetilde{s}})}$ where, $dist(a, b) = (a^\top b - t_2)/t_1$ is the shifted and scaled cosine similarity, $t_1, t_2$ being constants. The first loss term $\ell_a$ penalizes the model if the value of the attribute $a$ for the manipulated object is different from the desired value $s_a^*$ in terms of probabilities. The second term $\ell_{\overline{a}}$, on the other hand, penalizes the model if the values of any of the other attributes $a'$, deviate from their original values. Apart from these losses, we also include following additional losses.

$$\ell_{\text{cycle}} = \|o - g_a(\widetilde{o}; c_{\text{old}})\|_2 \tag{3}$$

$$\ell_{\text{consistency}} = \|o - g_a(o; c_{\text{old}})\|_2 \tag{4}$$

$$\ell_{\text{objGAN}} = -\sum\nolimits_{o' \in O} \left[\log D(o') + \log(1 - D\left(g_a(o'; c)\right))\right] \tag{5}$$

where $c_{old}$ is the original value of the attribute $a$ of object $o$, before undergoing change. Intuitively the first loss term $\ell_{\text{cycle}}$ says that changing an object and then changing it back should result in the same object. The second loss term $\ell_{\text{consistency}}$ intuitively means that changing an object $o$ that has value $c_{old}$ for attribute $a$, into a new object with the same value $c_{old}$, should not result in any change. These additional losses prevent the change network from changing attributes which are not explicitly taken care in earlier losses (1) and (2). For example, rotation or location attributes of the objects that are not part of our DSL. We also impose an adversarial loss $\ell_{\text{objGAN}}$ to ensure that the new object embedding $\widetilde{o}$ is from the same distribution as real object embeddings. $D$ stands for discriminator in equation (5). Total loss is a weighted sum of equations (1) to (5).

| Method | $\beta = 5.4k$ | | | $\beta = 7k$ | | | $\beta = 10k$ | | | $\beta = 20k$ | | | $\beta = 54k$ | | |
|---|---|---|---|---|---|---|---|---|---|---|---|---|---|---|---|
| | FID | $R1$ | $R3$ | FID | $R1$ | $R3$ | FID | $R1$ | $R3$ | FID | $R1$ | $R3$ | FID | $R1$ | $R3$ |
| GeNeVA | 27.6 | 5.9 | 36.3 | – | – | – | – | – | – | – | – | – | 16.0 | 4.1 | 39.4 |
| TIM-GAN | 18.0 | 41.0 | 72.1 | **15.0** | 42.9 | 73.5 | **13.1** | 49.8 | 77.3 | **14.8** | **62.5** | **84.2** | **13.5** | **78.3** | **92.3** |
| NEUROSIM | **16.6** | **57.2** | **79.4** | 16.4 | **57.3** | **79.3** | 16.9 | **57.2** | **79.3** | 16.7 | 57.2 | 79.4 | 16.8 | 57.1 | 79.3 |

Table 1: Performance comparison of NEUROSIM with TIM-GAN [13] and GeNeVA [3] with varying the number of examples ($\beta$) of CIM-NLI used during training. The '-' entries for GeNeVa were not computed due to excessive training time; it's performance is low even when using full data.

### 3.3 Image Rendering from Scene Graph

**5) Rendering Network:** Design and training methodology for this module closely follows [9]. We take multiple images $\{I_1, I_2 \cdots I_n\}$ and generate their scene graph $\{G_{I_1}, G_{I_2} \cdots G_{I_n}\}$ using the *visual representation network* described earlier. Each of these scene graphs is then transformed into the final image using the scene graph to image generation architecture described in [9].

## 4 Experiments

**1) Dataset:** We create a *multi-object multi-hop instruction* based image manipulation dataset, referred to as CIM-NLI. This dataset was generated with the help of CLEVR toolkit [8]. CIM-NLI consists of *(source image $I$, manipulation instruction text $T$, manipulated target image $\widetilde{I^*}$)* tuples.

**2) Baselines:** We compare our model with supervised approaches, TIM-GAN[13] and GeNeVA[3].

**3) Evaluation Metrics:**

For evaluation on image manipulation task, we use two metrics - i) FID, ii) Recall@$k$. FID [5] measures the realism of the generated image. Recall@$k$ measures the semantic similarity of gold manipulated image $\widetilde{I}^*$ and system produced manipulated image $\widetilde{I}$. We compute Recall@$k$ as in Zhang et al. [13].

| Method | Reasoning | |
| --- | --- | --- |
| | $ZH$ | $MH$ |
| GeNeVA (5.4$K$) | 4.7 | 6.3 |
| TIM-GAN (5.4$K$) | 54.5 | 36.5 |
| NEUROSIM (5.4$K$) | **59.5** | **56.5** |

Table 2: $R1$ results for 0-hop (ZH) vs. multi-hop (MH) instruction guided image manipulation. Along with each method, number of training data points from CIM-NLI used are written in bracket.

**4) Results:** Let $\beta$ denote the number of examples from CIM-NLI dataset seen by NEUROSIM during training. Table 1 shows NEUROSIM (being weakly supervised) performs significantly better than baselines when trained with $\beta \leq 10k$ and very close to its closest competitor with $\beta = 20k$ examples using the $R@1$ performance metric. This clearly demonstrates the strength of NEUROSIM in learning to manipulate while only making use of VQA annotations.

**5) Multi-hop Reasoning Performance:** Table 2 compares baselines with NEUROSIM on instructions requiring 0-hop versus multi-hop (1-3 hops) reasoning. When dealing with multi-hop instructions performance of TIM-GAN drops significantly, whereas NEUROSIM results in almost equal performance on both sets implying that it is effective at handling complex reasoning tasks.

**6) Qualitative Assessment:** Figure 3 compares the images generated by NEUROSIM, TIM-GAN, and GeNeVA. Some of the images generated by TIM-GAN and GeNeVA contain smudges where the manipulation has been done partially, while NEUROSIM suffers less from this problem. Overall visual appearance of TIM-GAN is better in a some cases owing to fully supervised training.

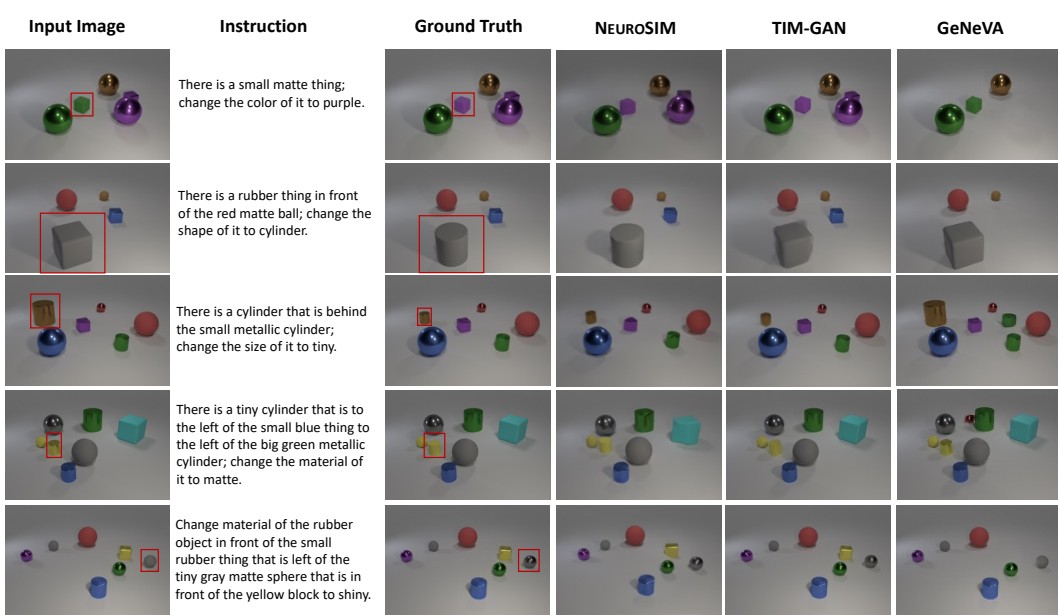

Figure 3: Visual comparison of NEUROSIM results with TIM-GAN [13] and GeNeVA [3].

## 5 Conclusion and Future Work

In this paper, we present a neuro-symbolic approach NEUROSIM to solve image manipulation task using weak supervision of VQA annotations. Our approach builds on existing work on neuro-symbolic VQA [10] to incorporate manipulation operations. The manipulation is achieved via generation of latent symbolic programs. Our experiments on a newly created dataset of image manipulation demonstrates the potential of our approach compared to supervised baselines. In future, we intend to extend this approach to other image editing operators and real images.

## Acknowledgements

This work was supported by an IBM AI Horizons Network (AIHN) grant. We thank IIT Delhi HPC facility[3], IBM cloud facility, and IBM Cognitive Computing Cluster (CCC) for computational resources. We thank anonymous reviewers for their insightful comments that helped in further improving our paper. Parag Singla was supported by the DARPA Explainable Artificial Intelligence (XAI) Program with number N66001-17-2-4032, IBM AI Horizon Networks (AIHN) grant and IBM SUR awards. Any opinions, findings, conclusions or recommendations expressed in this paper are those of the authors and do not necessarily reflect the views or official policies, either expressed or implied, of the funding agencies.

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

# Image Manipulation via Neuro-Symbolic Networks (Appendix)

## A Domain Specific Language (DSL)

Table 1 captures the DSL used by our NEUROSIM pipeline. The first 5 constructs in this table are common with the DSL used in [3]. The last operation (`Change`) was added by us to allow for the manipulation operation. Table 2 show the type system used by the DSL in this work. It is inherited

| Operation | Signature [*Output ← Input*]) | Semantics |
|---|---|---|
| `Scene` | ObjSet ← () | Returns all objects in the scene. |
| `Filter` | ObjSet ← (ObjSet, ObjConcept) | Filter out a set of objects from ObjSet that have concept (e.g. red) specified in ObjConcept. |
| `Relate` | ObjSet ← (ObjSet, RelConcept, Obj) | Filter out a set of objects from ObjSet that have concept specified relation concept (e.g. RightOf) with object Obj. |
| `Query` | ObjConcept ← (Obj, Attribute) | Returns the Attribute value for the object Obj. |
| `Exist` | Bool ← (ObjSet) | Checks if the set ObjSet is empty. |
| `Change` | Obj ← (Obj, Concept) | Changes the attribute value of the input object (Obj), corresponding to the input concept, to Concept |

Table 1: Extended Domain Specific Language (DSL) used by NEUROSIM.

from [3].

| Type | Remarks |
|---|---|
| `ObjConcept` | Concepts for any given object, such as *blue, cylinder,* etc. |
| `Attribute` | Attributes for any given object, such as *color, shape,* etc. |
| `RelConcept` | Relational concepts for any given object pair, such as *RightOf, LeftOf,* etc. |
| `Object` | Depicts a single object |
| `ObjectSet` | Depicts multiple objects |

Table 2: Extended type system for the DSL used by NEUROSIM.

Neuro Causal and Symbolic AI Workshop at the 36th Conference on Neural Information Processing Systems (NeurIPS 2022).

## B  Dataset Details

### B.1  CIM-NLI Dataset

This dataset was generated with the help of CLEVR toolkit [2] by using following recipe.

1. First, we create a source image $I$ and the corresponding scene data by using *Blender* [1] software.
2. For each source image $I$ created above, we generate multiple instruction texts $T$'s using its scene data. These are generated using templates, similar to question templates proposed by [2].
3. For each such $(I, T)$ pair, we attach a corresponding symbolic program $P$ (not used by NEU-ROSIM though) as well as scene data for the corresponding changed image.
4. Finally, for each $(I, T)$ pair, we generate the target gold image $\widetilde{I}^*$ using Blender software and its scene data from previous step.

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
