# OpenReview forum: "Image Manipulation via Neuro-Symbolic Networks"
_NeurIPS.cc/2022/Workshop/nCSI — nCSI WS @ NeurIPS 2022 Poster_

### Official Review · Reviewer_EYB7 · 2022-10-14
**Interesting idea for image manipulation based on NS-CL in a paper with a multitude of minor flaws**

**Rating:** 2
**Confidence:** 2

**Review:**

**Summary of the Paper:**
In this paper, the authors extend NS-CL by an image manipulation task. The result is what they refer to as **NeuroSIM**, a model which uses visual representations, semantic parsing, and symbolic operations to allow for changes in a given source image in accordance with the input instruction text. In comparison to the TIM-GAN baseline, NeuroSIM performs better using a small number of training examples but does not improve significantly when trained on larger datasets, in which case TIM-GAN has the best results.

**Strengths:**
This paper clearly fits the topic of this workshop well, as it utilizes a neuro-symbolic approach in order to apply image manipulation. I like the underlying idea of the paper of extending NS-CL in such a way that the input image is not only looked at (to answer questions) but actively changed. As a personal thought (this is not claimed by the authors), this approach might possibly even be see as a step towards causality, going from only observing an image to *intervening* on it. The paper makes the differences to NS-CL and other related work very clear while still introducing the underlying concepts in sufficient detail. I also really like the examples given for some functions within NeuroSIM, which make understanding easier and faster. The figures are well made and helpful.

**Weaknesses/Possible Improvements:**
I would have liked to see at least some more experiments going beyond the comparison of metrics with the two baselines. For NS-CL, the authors show generalization abilities of their model in different aspects. With that being (as I see it) one of the big advantages of using symbolic models, I would have liked to see at least one experiment exploring this direction (see the NS-CL paper). Speaking of the results, I feel like the authors chose to focus a bit too much on the advantageous parts of their approach. They show that NeuroSIM performs better than TIM-GAN even on a relatively small ($\beta = 5.4k$) dataset (a good result worth highlighting), mention the almost-equal performance on a larger ($\beta = 20k$) dataset (a fair assessment) but fail to mention that NeuroSIM has worse results than TIM-GAN when trained with an even larger ($\beta = 54k$) dataset. Based on their experiments, NeuroSIM does not improve noticeably with larger datasets, indicating that from a specific dataset size on, TIM-GAN seems to be the preferable model (judging by FID and recall metrics). This also makes me question the meaningfulness of Table 2, where results are only calculated for a small dataset. Does NeuroSIM still come out on top when using larger datasets here? The same applies for the qualitative assessment, where the dataset size is not even mentioned. What dataset size is used here? NeuroSIM appears to produce better images here but would this still be true if the best TIM-GAN model was used (and if that was used, why not mention it)? Overall, I find the results and its discussion lacking and slightly biased towards the desirable results. Instead of discussing NeuroSIM by highlighting the beneficial properties (requiring less data, generalization abilities), the disadvantages of the model go unmentioned.

I do find the paper generally well-written and also like the use and design of figures. Nonetheless, it would have benefited from a lot more polishing. None of the following points is a significant problem in and of itself but they add up to leave a bad impression (some points might be slightly subjective, but most points simply are mistakes):

- The abstract seems too long for a 5 page paper. In my opinion, there is too much detail here, taking up space which could be used later (for example in the experiment section)
- Line 61: "s" missing for "task" (change "task" to "tasks"), also what are the "two cases"?
- Lines 122 to 126: The order/wording of sentences is confusing. An example is introduced before the definition.
- Line 134: unnecessary comma
- Equation 4: What is $D$? Discriminator? Should be mentioned. And what is the concept $c$ here?
- Table 1: It would be nice to highlight (**bold**) the best results per datset size/metric.
- Lines 149, 152: each missing a dot (full stop)
- Line 170: missing comma on line end
- Lines 172-173: Where is 6)? Why is it skipped?
- Generally, the punctuation surrounding equations does not seem right to me, in its current form it disrupts the reading flow

Note, that I did not specifically pay attention to spelling or punctuation errors, so going over the entire paper again could be beneficial.
I wonder why NeuroSIM despite the used losses visibly changes the appearance of other objects a bit (see the appendix). Should the corresponding loss be weighted more strongly?

**Review Summary:**
Overall, I see this paper as introducing a good idea in a nice, easy to understand way with results showing that it can work. However, a lacking experiment section and a general feeling of "unpolishedness" leave a bad impression. I decided to **barely recommend accepting** this paper despite its many flaws, since I like the underlying idea and methodology and since all of my personal complaints are fixable and do not harm the basic concept. I do so in good faith, expecting that the authors will decide to polish the paper for the final version and hope to see at least some changes made to the experiments section.

---

### Official Review · Reviewer_6tK3 · 2022-10-15

**Rating:** 2
**Confidence:** 1

**Review:**


The paper proposes a framework for image manipulation via natural language text using a neuro-symbolic approach. The authors propose NeuroSIM, an image manipulation model based on the Neuro-Symbolic Concept Learner (NSCL).

$Pros$
- The authors propose changing the source image by manipulating its scene representation graph given an instruction parsed from the natural language. The devised Manipulation Network adapts the scene graph without the need to work directly on the image.
- The results of the NeuroSIM model seem to be especially competitive in the low data regime, which may be a desired property if access to larger datasets is impossible. The proposed method seems to also work better for multi-hop instructions.

$Cons$
-  The results from the multi-hop instructions are interesting; however,  they could have also been better investigated – i.e., Similarly to the results from Table 1, what would be the outcome in a higher data regime (larger beta?). Why is only R1 reported in this table?
-  The Description of the training of the Semantic Pruning Module, Visual Representation Network, and Concept Quantization Network is very vague. Understandably, the authors do not want to waste space in the main text to explain methods adapted from some other work. However, some more detailed explanations could be included in the appendix. It would make the reading more accessible, and it is important to understand the input of the three abovementioned components since the proposed framework relies heavily on them. Have the components been adapted from a trained NSCL module or trained/fine-tuned with other parts of NeuroSIM?

I would like to state that even though I find this work quite interesting, it is outside of the scope of my everyday research, and hence I am not very familiar with the subject of the work and the problems it is trying to solve.

---

### Meta-Review · Area_Chair_pByX · 2022-10-19

**Recommendation:** 2
**Confidence:** 3

**Metareview:**

The paper discusses a NeSy approach for VQA. The reviewers found the paper generally well written, but somewhat lacking in rigorous comparisons. The second reviewer in particular has identified a number of areas of improvement. I don't know if the authors will be able to implement all the suggested changes by the camera ready deadline, but I'd certainly strongly recommend that they at least try. For the workshop the fit is good (but see below), but they should explore the leveraging of the symbolic angle is a bit more comprehensively as suggested by R2.

PS I also suggest they take a more careful look at the connection to causality. As R1 suggests manipulation could be seen as interventions, but I don’t wish to see the reviewer make the connection. This workshop is about causal + symbolic, not just plain neurosymbolic, so some discussion on the causal connection is warranted. Thus, weak accept.

---

### Decision · Program_Chairs · 2022-10-20

Accept (Poster)